

# Three-dimensional ionospheric conductivity associated with pulsating auroral patches: Reconstruction from ground-based optical observations

Mizuki Fukizawa[1], Yoshimasa Tanaka[1, 2, 3], Yasunobu Ogawa[1, 2, 3], Keisuke Hosokawa[4], Tero Raita[5]

[1]National Institute of Polar Research, Tachikawa, 190-8518, Japan
[2]Polar Environment Data Science Center, Joint Support-Center for Data Science Research, Research Organization of Information and Systems, Tachikawa, 190-0014, Japan
[3]Department of Polar Science, The Graduate University for Advanced Studies (SOKENDAI), Tachikawa, 190-8518, Japan
[4]Graduate School of Informatics and Engineering, University of Electro-Communications, Chofu, 182-8585, Japan
[5]Sodankylä Geophysical Observatory, University of Oulu, Oulu, FI-90014, Finland

*Correspondence to*: Mizuki Fukizawa (fukizawa.mizuki@nipr.ac.jp)

**Abstract.** Pulsating auroras (PsAs) appear over a wide area within the aurora oval from the midnight sector to the noon sector. In previous studies, observations by magnetometers onboard satellites have reported the presence of field-aligned currents (FACs) near the edges and interiors of pulsating aurora patches. PsAs are thus a key research target for understanding the magnetosphere–ionosphere coupling process. However, the three-dimensional (3-D) structure of the electric currents has yet to be clarified, since each satellite observation is limited to the single dimension along its orbit. This study's aim was a reconstruction of the 3-D structure of ionospheric conductivity, which is necessary to elucidate the 3-D ionospheric current. Tomographic analysis was used to estimate the 3-D ionospheric conductivity for rapidly changing auroral phenomena such as PsAs. The reconstructed Hall conductivity reached its maximum value of $1.4 \times 10^{-3}$ S m$^{-1}$ at 94 km altitude, while the Pedersen conductivity reached its maximum value of $2.6 \times 10^{-4}$ S m$^{-1}$ at 116 km altitude. The Pedersen conductivity, which is driven by the motion of electrons, exhibited a secondary peak value of $9.9 \times 10^{-5}$ S m$^{-1}$ at 86 km altitude. The electron Pedersen conductivity maximum value in the *D* region was approximately 38% of the ion Pedersen conductivity maximum value in the *E* region. The FAC, derived under the assumption of a uniform ionospheric electric field, was approximately 70 µA m$^{-2}$ near the edge of the PsA patch. This FAC value was approximately 10 times that observed by satellites in previous studies. If the conductivity around the patch is underestimated or the assumption of a uniform field distribution is incorrect, the FAC could be overestimated. On the contrary, due to sharper boundary structures, the FAC could actually have had such a large FAC.

## 1 Introduction

Pulsating auroras (PsAs) are a kind of diffuse aurora with quasi-periodic luminosity modulation of ~2–20 s (Yamamoto, 1988). Although PsAs are dimmer in brightness than typical discrete auroras (some hundreds of Rayleigh (R) to tens of kR at



OI 557.7 nm; a few hundred R to ~10 kR at $N_2^+$ first negative band 427.8 nm) (McEwen et al., 1981; Royrvik and Davis, 1977), they are a general auroral phenomenon because of their wide range of appearance from the midnight to noon sectors depending on geomagnetic activities (Oguti et al., 1981; Royrvik and Davis, 1977). Typically, PsAs are classified into three types: auroral arcs, arc segments, and patches. This study focuses on patch-type PsAs.

Previous observations by satellite-borne magnetometers have indicated the occurrence of field-aligned currents (FACs) associated with PsA patches (Fujii et al., 1985; Gillies et al., 2015). Pairs of FACs flowing into and out of PsA patch edges are thought to be closed via the Pedersen current flowing within the PsA patch in the direction perpendicular to the geomagnetic field (Fujii et al., 1985; Oguti et al., 1984). PsAs are caused by precipitating electrons from a few keV to tens of keV scattered into the loss cone by whistler-mode chorus waves excited near the magnetic equator (Kasahara et al., 2018;
Nishimura et al., 2010, 2011). Chorus waves, which propagate to the off-equator, scatter high-energy electrons from tens of keV to a few MeV and enhance the electron density in the ionospheric $D$ region (Kawamura et al., 2021; Miyoshi et al., 2010, 2015, 2020, 2021; Shumko et al., 2021). Hosokawa and Ogawa (2010) investigated the altitude profile of the electron density obtained with the European Incoherent Scatter (EISCAT) radar, reporting that the Pedersen current carried by electrons was seen in the $D$ region (at approximately 80–95 km altitudes) associated with PsAs in addition to the commonly
seen ion Pedersen current in the $E$ region (at an altitude of 120 km). They suggested that the electron Pedersen current in the $D$ region contributed to FAC closure in the ionosphere because the PsA emission altitude was closer to the peak altitude of the electron Pedersen current layer than the ion Pedersen layer, even though the Pedersen conductivity in the $D$ region was only $\approx 8 \times 10^{-5}$ S m$^{-1}$, a mere 13% that in the $E$ region ($6 \times 10^{-4}$ S m$^{-1}$).

As mentioned above, the electric current structure associated with PsAs has been extensively studied by satellite and
EISCAT radar observations. However, such observations are limited to the one dimension along the satellite's orbit or radar's beam direction. This is why the three-dimensional (3-D) structure of the current system has not been clarified. To elucidate the 3-D ionospheric currents, it is necessary to know the 3-D ionospheric conductivity and electric field. The 3-D distribution of the ionospheric electric field will be obtained from the 3-D ion velocity vectors observed by the EISCAT_3D radar (https://eiscat.se/) (Stamm et al., 2023), which will begin observations in late 2023. On the contrary, it is difficult to
obtain 3-D ionospheric conductivity measurement with high temporal and spatial resolutions for auroral phenomena given their high spatiotemporal variability, as seen in PsAs in particular.

This study aims to reconstruct the 3-D structure of the ionospheric conductivity associated with PsAs using computed tomographic analysis, a useful analysis method for measuring 3-D ionospheric physical quantities (Fukizawa et al., 2022). Generalized-Aurora Computed Tomography (G-ACT) is a method of reconstructing the two-dimensional electron flux and
3-D volume emission rate from monochromatic auroral images obtained with all-sky cameras (ASCs) at multiple locations (Aso et al., 2008; Tanaka et al., 2011). Fukizawa et al. (2022) demonstrated that the electron density altitude profile observed by the EISCAT radar could be reconstructed correctly using G-ACT. In this study, the 3-D Pedersen and Hall conductivities are reconstructed by combining a neutral atmosphere model and the 3-D electron density reconstructed by G-ACT.



## 2 Methods and Observations

### 2.1 Derivation of the three-dimensional ionospheric conductivity

The Pedersen conductivity $\sigma_\mathrm{P}$ and Hall conductivity $\sigma_\mathrm{H}$ can be written respectively as (Jones, 1974)

$$\sigma_\mathrm{P} = \frac{en_\mathrm{e}}{B}\left(\frac{\Omega_\mathrm{e}\nu_\mathrm{en}}{\Omega_\mathrm{e}^2+\nu_\mathrm{en}^2} + \frac{\Omega_\mathrm{i}\nu_\mathrm{in}}{\Omega_\mathrm{i}^2+\nu_\mathrm{in}^2}\right), \tag{1}$$

$$\sigma_\mathrm{H} = \frac{en_\mathrm{e}}{B}\left(\frac{\Omega_\mathrm{e}^2}{\Omega_\mathrm{e}^2+\nu_\mathrm{en}^2} - \frac{\Omega_\mathrm{i}^2}{\Omega_\mathrm{i}^2+\nu_\mathrm{in}^2}\right), \tag{2}$$

where $e$ [C] is the elementary charge, $n_\mathrm{e}$ [m$^{-3}$] is the electron density, $\Omega_\mathrm{i}$ and $\Omega_\mathrm{e}$ [rad s$^{-1}$] are the ion and electron gyrofrequencies, respectively, and $\nu_\mathrm{in}$ and $\nu_\mathrm{en}$ [s$^{-1}$] are the ion-neutral and electron-neutral collision frequencies, respectively. $\nu_\mathrm{in}$ and $\nu_\mathrm{en}$ were estimated using the following equations (Brekke, 2013):

$$\nu_\mathrm{in} = 4.34 \times 10^{-16} n(\mathrm{N}_2) + 4.28 \times 10^{-16} n(\mathrm{O}_2) + 2.44 \times 10^{-16} n(\mathrm{O}), \tag{3}$$

$$\nu_\mathrm{en} = 5.4 \times 10^{-16}[n(\mathrm{N}_2) + n(\mathrm{O}_2) + n(\mathrm{O})] \cdot \sqrt{T_\mathrm{e}}. \tag{4}$$

Here, $n(\mathrm{N}_2)$, $n(\mathrm{O}_2)$, and $n(\mathrm{O})$ [cm$^{-3}$] are the ionospheric densities of nitrogen molecules, oxygen molecules, and oxygen atoms, respectively, and $T_\mathrm{e}$ [K] is the electron temperature. We assumed that the electron temperature could be approximated by the neutral temperature in the altitude range considered in this study when the electric field was not very high (<30 mV m$^{-1}$) (Hosokawa and Ogawa, 2010). The neutral densities and temperature were taken from the Mass-Spectrometer-Incoherent-Scatter (MSIS)-E00 model (Picone et al., 2002).

### 2.2 Derivation of the three-dimensional electron density distribution

We reconstructed the 3-D distribution of $n_\mathrm{e}$ in Eqs. (1) and (2) from auroral images using G-ACT, which reconstructs the electron flux from observed auroral images by maximizing the following posterior probability based on Bayes' theorem (Tanaka et al., 2011):

$$P(\boldsymbol{f}|\widetilde{\boldsymbol{g}}) \propto \exp\left[-\frac{1}{2}\left\{\left(\widetilde{\boldsymbol{g}} - \boldsymbol{g}(\boldsymbol{f})\right)^T \Sigma^{-1}\left(\widetilde{\boldsymbol{g}} - \boldsymbol{g}(\boldsymbol{f})\right) + \frac{\left\|\nabla^2 \boldsymbol{f}\right\|^2}{\sigma^2}\right\}\right]. \tag{5}$$

where $\boldsymbol{f}$ [m$^{-2}$ s$^{-1}$ eV$^{-1}$] is the electron flux; $\widetilde{\boldsymbol{g}}$ [R] is the gray level of the observed auroral image; $\boldsymbol{g}(\boldsymbol{f})$ [R] is the grey level of the pseudo auroral image; $\Sigma^{-1}$ is the inverse covariance matrix; $\nabla^2 \boldsymbol{f}$ is the second-order derivative of $\boldsymbol{f}$ with respect to $x$, $y$, and $E$; and $\sigma^2$ is the variance of $\nabla^2 \boldsymbol{f}$. Here, we assumed that the pixel values in auroral images are mutually independent. In this case, the covariance matrix $\Sigma$ is a diagonal matrix (i.e., the off-diagonal terms are zero) whose non-zero elements are the variance of each pixel value. The variance was determined from auroral images obtained with each ASC from 01:00 to 02:00 UT on February 18, 2018. The pseudo auroral image $\boldsymbol{g}(\boldsymbol{f})$ was obtained by integrating the 3-D volume emission rate along the line-of-sight direction from each pixel of each auroral image (Eq. (7) in Tanaka et al. (2011)). The 427.8-nm and





557.7-nm volume emission rates were derived from the electron flux using the Global Airglow (GLOW) model (Solomon, 2017). The monoenergetic electron flux was specified in the GLOW model. We maximized $P(\boldsymbol{f}|\tilde{\boldsymbol{g}})$ in Eq. (5) by minimizing the following function:

$$\varphi(\boldsymbol{f}; \lambda, c_j) = \sum_j \left(c_j\tilde{\boldsymbol{g}}_j - \boldsymbol{g}_j(\boldsymbol{f})\right)^T \boldsymbol{\Sigma}_j^{-1} \left(c_j\tilde{\boldsymbol{g}}_j - \boldsymbol{g}_j(\boldsymbol{f})\right) + \lambda^2\|\nabla^2\boldsymbol{f}\|^2 = \|\boldsymbol{r}(\boldsymbol{f}; \lambda, c_j)\|^2, \qquad (6)$$

where

$$\boldsymbol{r}(\boldsymbol{f}; \lambda, c_j) = \begin{pmatrix} \boldsymbol{\Sigma}_j^{-\frac{1}{2}}\left(c_j\tilde{\boldsymbol{g}}_j - \boldsymbol{g}_j(\boldsymbol{f})\right) \\ \lambda\nabla^2\boldsymbol{f} \end{pmatrix}, \qquad (7)$$

$\lambda$ is the weighting factor for the spatial and energy derivative terms, and $c_j$ is the correction factor for the relative sensitivity among ASCs. The subscript $j$ is ian index representing the various observation points. The method to determine these parameters is explained in Section 2.4.

We carried out the change of variables $\boldsymbol{f} = \exp(\boldsymbol{x})$ to take advantage of the non-negative constraint on the electron flux $\boldsymbol{f}$ (i.e., $\boldsymbol{f} \geq 0$). The initial value was obtained by minimizing the function $\varphi(\boldsymbol{x}; \lambda, c_j)$ by the Simultaneous Iterative Reconstruction Technique method (Gordon et al., 1970; Tanabe, 1971). We then minimized the function $\varphi(\boldsymbol{x}; \lambda, c_j)$ by implementing the Gauss–Newton (GN) method to reconstruct the electron flux and volume emission rate. In the GN method, the parameter $\boldsymbol{x}$ was iterated according to $\boldsymbol{x}^{(k+1)} = \boldsymbol{x}^{(k)} + \Delta\boldsymbol{x}^{(k)}$. The step size at the $k$-th step was determined by solving the

equation

$$\left(\boldsymbol{J}^T(\boldsymbol{x}^{(k)})\boldsymbol{J}(\boldsymbol{x}^{(k)})\right)\Delta\boldsymbol{x}^{(k)} = -\boldsymbol{J}^T(\boldsymbol{x}^{(k)})\boldsymbol{r}(\boldsymbol{x}^{(k)}), \qquad (8)$$

where $\boldsymbol{J}(\boldsymbol{x})$ is the Jacobian matrix of $\boldsymbol{r}(\boldsymbol{x})$ with respect to $\boldsymbol{x}$. We solved Eq. (8) by the Conjugate Gradient (CG) method. The number of iterations of the CG method was set to 20. The criterion to stop the GN method was $\sqrt{\|\Delta\boldsymbol{x}^{(k)}/\boldsymbol{x}^{(k)}\|^2} < 10^{-2}$.

The reconstructed volume emission rate was converted to the electron density using the continuity equation of the electron

density (Eq. (4) in Fukizawa et al. (2022)), i.e.,

$$\frac{\partial n_e}{\partial t} = kL - \alpha_{\mathrm{eff}}n_e^2, \qquad (9)$$

where $n_e$ [m$^{-3}$] is the electron density, $L$ [m$^{-3}$ s$^{-1}$] is the volume emission rate, $k$ is a positive constant for converting the volume emission rate to the ionization rate (see Appendix B in Fukizawa et al. (2022)), and $\alpha_{\mathrm{eff}}$ [m$^3$ s$^{-1}$] is the effective recombination rate. Eq. (9) was solved with the Runge–Kutta method. Fukizawa et al. (2022) confirmed that the

reconstructed electron density was almost perfectly consistent with that observed by the European Incoherent Scatter (EISCAT) radar.





The origin of the simulation region was 69.4°N and 19.2°E. The *z*-axis was antiparallel to the geomagnetic field, the *x*-axis was antiparallel to the horizontal component of the geomagnetic field, and the *y*-axis was parallel eastward (see Fig. 2 in Tanaka et al. (2011)). The simulation region ranged from −75 to 75 km, from −100 to 100 km, and from 80 to 180 km for the

*x*, *y*, and *z* axes, respectively. The spatial resolution was 2 × 2 × 2 km.

## 2.3 Optical observations

We used monochromatic auroral images obtained with ASCs at 4 ground-based stations: Abisko (ABK: 68.36°N, 18.82°E), Kilpisjärvi (KIL: 69.05°N, 20.78°E), Skibotn (SKB: 69.35°N, 20.36°E), and Tromsø (TRO: 69.58°N, 19.22°E). ASCs to observe 427.8-nm auroral emission were installed at ABK, KIL, and SKB, while those for 557.7 nm were at KIL, SKB, and

TRO. The field of view of each ASC at an altitude of 100 km is shown in Figure 1. ASCs for 427.8 nm at ABK and KIL were from the aurora observation network called Magnetometers Ionospheric Radars All-sky Cameras Large Experiment (MIRACLE) (Sangalli et al., 2011). The filter wavelength attached to the MIRACLE ASCs is 427.8±2 nm. The other ASCs were Watec Monochromatic Imagers (WMIs) (Ogawa et al., 2020). The filter wavelengths of WMIs were $430 \pm 5$ and $560 \pm 5$ nm. The pixel numbers were $512 \times 512$ pixels for MIRACLE ASCs and $640 \times 480$ pixels for WMIs. The exposure

time was 1 s to observe 557.7-nm emission, but 2 s for dimmer 427.8-nm emission. The 557.7 nm auroral images were integrated every 2 s to adjust the time resolution of the 427.8-nm auroral images. To improve the signal-to-noise ratio, we composited 427.8-nm auroral images obtained from four same-type WMIs at SKB. In addition, the median filter of $3 \times 3$ pixels was applied to all auroral images. When the same median filter was applied to dark images subtracted from the auroral images, unexpected small structures appeared in the subtracted auroral images. To avoid this over-subtraction, we applied a

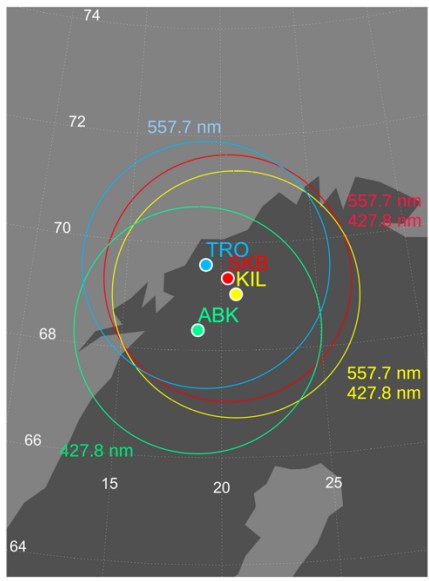

**Figure 1.** The field of views of all-sky cameras in Abisko (ABK), Kilpisjärvi (KIL), Skibotn (SKB), and Tromsø (TRO) at an altitude of 100 km.





$7 \times 7$ pixel median filter for the dark images. After the subtraction of the dark images, we conducted the flat field correction, Gamma correction, and conversion from count to Rayleighs. The relationship between count and Rayleighs can be written as the following equations.

$$[\text{Count}'] = 255 \times ([\text{Count}]/255)^{(1/\gamma)} \quad (10)$$

$$[\text{Rayleighs}] = a \times [\text{Count}'] + b \quad (11)$$

Here, Count is raw count, Count' is the Gamma-corrected count, $\gamma$ is the Gamma value, and $a$ and $b$ are constants. The parameters are summarized in Table 1.

Figures 2a–f show, at each station and each wavelength, the keogram, which is the time series of the auroral image sliced along the magnetic latitude to pass through the pixel containing the EISCAT radar observation point from 00:30 to 01:20 UT on February 18, 2018. After an auroral breakup at 00:10 UT, diffuse auroras, PsAs, and auroral streamers were observed by

ASCs at stations except for KIL (Video A1). It was cloudy at KIL until 00:47 UT. A PsA patch was detected at the EISCAT radar observation pixel at 00:53:30–00:53:42 UT, as shown by black arrows in Figure 2. The electron density increased in association with the PsA patch even below 85 km altitude (Figure 2g). As shown in Figure 3, this PsA patch was classified as an expanding PsA, which expands from a core and then recedes back.

The PsA emissions in the auroral images were embedded in emissions such as diffuse aurora, sunlight, moonlight, and city

light (Figures 4a–b). These background emissions, which have no specific structure, should be subtracted before conducting G-ACT because they cause ambiguity in the reconstruction results. The background emission image $\boldsymbol{g}_{\text{bk}}$ was defined as

$$\boldsymbol{g}_{\text{bk}} = c\boldsymbol{g}_0 + d, \quad (12)$$

where $\boldsymbol{g}_0$ [R] is a pseudo image derived by assuming all voxels had the same volume emission rate of 1 cm$^{-3}$ s$^{-1}$, $d = \min(\widetilde{\boldsymbol{g}}) - \min(c\boldsymbol{g}_0)$, and $c$ is a constant whose value is chosen to minimize $|\widetilde{\boldsymbol{g}} - \boldsymbol{g}_{\text{bk}}|^2$. We eliminated the gray level at

pixels whose zenith angle was larger than 80°. The determined background emission profiles are shown with orange lines in Figure 4b. The values of $c$ and $d$ in Eq. (12) differ because the zenith angle of the sun or moon and the brightness of city light depend on the position of the observation point (Table 1). These values also varied with time due to temporal variation in diffuse auroral emission intensity. Figure 4c shows auroral images with background emission subtracted.

Although the absolute value of the auroral image from each ASC has been corrected by calibration experiments in a

laboratory, the images from different ASCs had different absolute values during actual observations due to differences in the sensitivity degradation and transmission of acrylic domes at each station. These differences were corrected by the parameter $c_j$ in Eq. (6). The method to determine this parameter is explained in the next section.





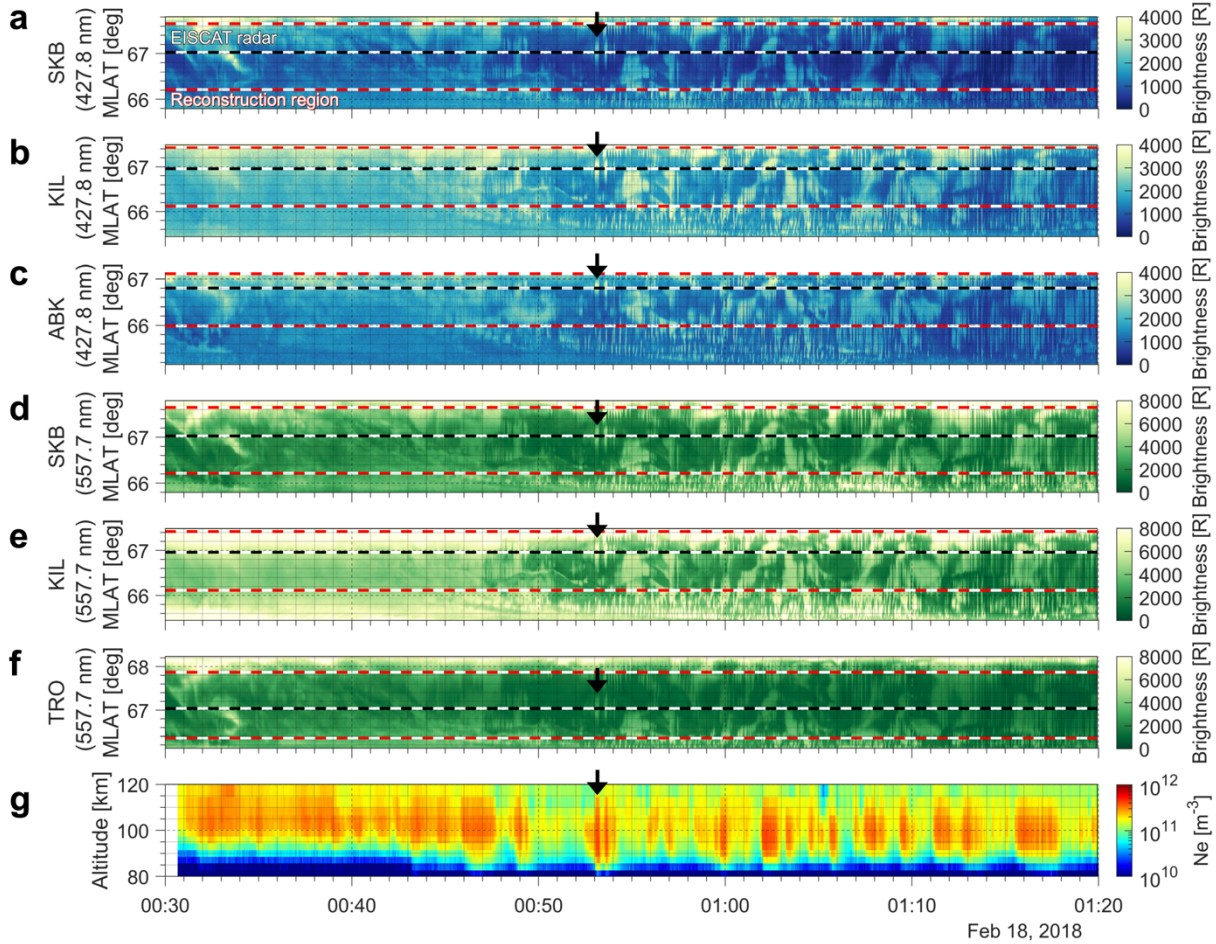

**Figure 2. (a–f)** Keogram (time series of auroral image sliced along the magnetic latitude (MLAT) to pass through the pixel containing the EISCAT radar observation point) at each station and each wavelength, from 00:30 to 01:20 UT on February 18, 2018. **(g)** The electron density observed by the EISCAT radar in Tromsø. Black and red dashed lines show the EISCAT radar observation point and reconstruction region, respectively. Black arrows indicate the pulsating auroral patch analyzed in this study.

**Table 1.** Filter wavelength and constants in Eqs. (10), (11), and (12) for each camera. $c$ and $d$ values are only for the 00:53:36 UT timepoint shown in Figure 4.

| Camera | Station | Filter wavelength (nm) | $\gamma$ | $a$ | $b$ | $c$ | $d$ |
|---|---|---|---|---|---|---|---|
| MIRACLE | ABK | $427.8 \pm 2$ | 1.00 | 4.68 | 0.00 | 70 | 173 |
| MIRACLE | KIL | $427.8 \pm 2$ | 1.00 | 4.03 | 0.00 | 105 | 157 |
| WMI | SKB | $430 \pm 5$ | 0.45 | 20.95 | −17.65 | 80 | −386 |
| WMI | KIL | $560 \pm 5$ | 0.45 | 21.63 | 118.85 | 425 | −2516 |
| WMI | SKB | $560 \pm 5$ | 0.45 | 21.63 | 118.85 | 230 | −1204 |
| WMI | TRO | $560 \pm 5$ | 0.45 | 21.63 | 118.85 | 200 | −974 |



**Figure 3.** 427.8-nm and 557.7-nm auroral images obtained from all-sky cameras in Abisko (ABK), Kilpisjärvi (KIL), Skibotn (SKB), and Tromsø (TRO) from 00:53:30 to 00:53:42 UT on February 18, 2018. The red plus and lines represent the EISCAT radar observation pixel and reconstruction region boundaries, respectively.



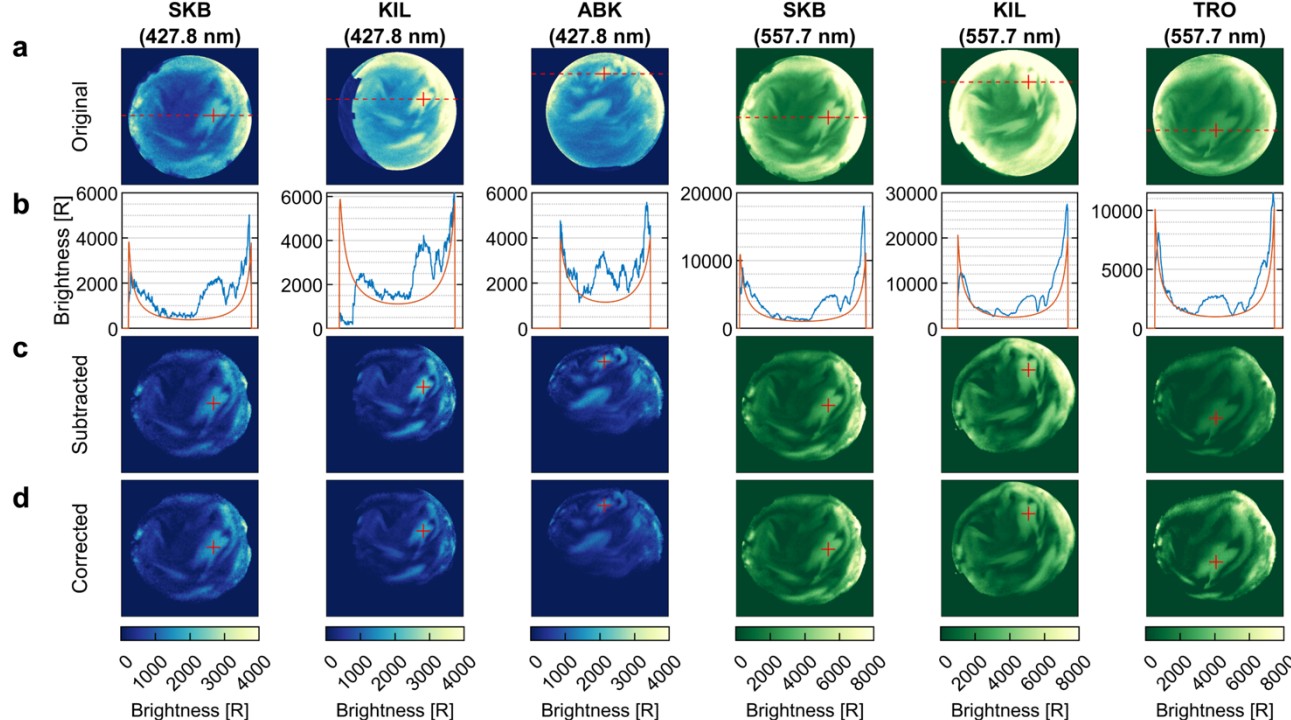

**Figure 4. (a)** Observed auroral images at 00:53:36 UT on February 18, 2018. Red pluses represent EISCAT radar observation pixels. **(b)** Brightness of observed auroral images (blue lines) and estimated background emission (orange lines) along red dashed lines in Figure 4a. **(c, d)** Auroral images with **(c)** background emission subtracted and **(d)** relative sensitivity corrected.


### 2.4 Determination of hyperparameters

Before conducting G-ACT, the hyperparameters $\lambda$ and $c_j$ in Eq. (6) must be determined. First, we determined $c_j$ ($j = 1, 2,\ldots,$ 6) with $\lambda$ fixed at $10^{-5}$ using fivefold cross-validation (Stone, 1974). Elements of the observed auroral image vector $\widetilde{\boldsymbol{g}}$ were divided into five subsets. Then, one subset was selected as the test set ($\widetilde{\boldsymbol{g}}_j^{\text{tes}}$) and the others as the training set ($\widetilde{\boldsymbol{g}}_j^{\text{tra}}$). We

found the solution $\hat{\boldsymbol{x}}$ to minimize $\varphi(\boldsymbol{x}; \lambda, c_j)$ using only the training set $\widetilde{\boldsymbol{g}}_j^{\text{tra}}$ and then predicted the test set $\boldsymbol{g}_j^{\text{tes}}(\hat{\boldsymbol{x}})$. We calculated the residual sum of squares between the actual and predicted values for the test data:

$$\delta(c_j) = \sum_j \left\| c_j \widetilde{\boldsymbol{g}}_j^{\text{tes}} - \boldsymbol{g}_j^{\text{tes}}(\hat{\boldsymbol{x}}) \right\|^2. \text{(13)}$$

The cross-validation score $\bar{\delta}(c_j)$ was calculated by averaging over five values of $\delta(c_j)$, which were obtained by using a different set as the training set each time. To save computational cost, cross-validation was performed separately for each

wavelength. The relative sensitivities of SKB were fixed to be 1 for each wavelength, while the other relative sensitivities were determined using the grid search method. The determined values were 0.78 for ABK (427.8 nm), 0.73 for KIL (427.8




nm), 0.61 for KIL (557.7 nm), and 1.92 for TRO (557.7 nm). Auroral images with relative sensitivity corrected are shown in Figure 4d.

The value of the parameter $\lambda$ was selected so as to minimize the difference between the electron densities observed by the EISCAT radar and those reconstructed by G-ACT. We conducted G-ACT for $\lambda = 10^{-5}, 10^{-4}, 10^{-3}, 10^{-2}, 10^{-1}$, and $10^{0}$ and found that the difference reached a minimum at $\lambda = 10^{-1}$.

**2.5 Validation of reconstruction accuracy using model aurora**

In Fukizawa et al. (2022), PsA patches were reconstructed using only 427.8-nm auroral images at three stations (ABK, KIL, and TRO). On the contrary, we improved the G-ACT method by adding auroral images at another wavelength (557.7 nm) at three stations (KIL, SKB, and TRO). The total number of stations was increased from three to four due to the addition of TRO. We performed G-ACT for a model aurora to determine the extent of reconstruction accuracy improvement associated with including the additional wavelength and observation point. First, horizontal distributions of total energy flux and average energy were prepared for three adjacent patches. The total energy flux was assumed to have a Gaussian distribution with a peak value of 1.6 mW m$^{-2}$ (Figure 5a). A uniform distribution with a value of 30 keV was assumed for the average

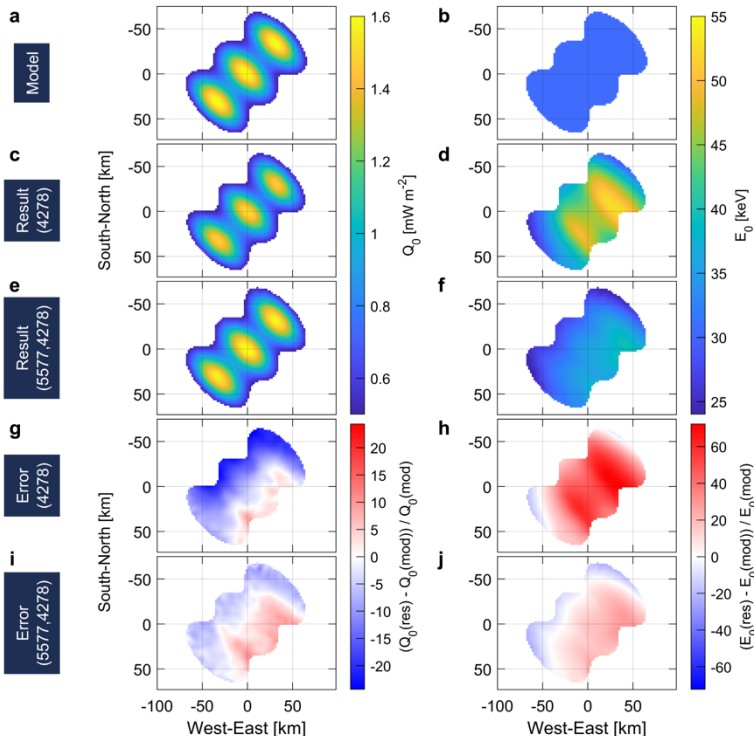

**Figure 5.** Models of **(a)** the total energy flux $Q_0$ and **(b)** average energy $E_0$. **(c–f)** Reconstructed results of **(c, e)** the total energy flux and **(d, f)** average energy using **(c, d)** only 427.8-nm auroral images or **(e, f)** both 557.7-nm and 427.8-nm auroral images. **(g–j)** Relative reconstruction errors calculated as (Result – Model) / Model.





**Table 2.** Errors of the total energy flux ($Q_0$) and average energy ($E_0$) reconstructed using only 427.8-nm auroral images vs. using both 557.7-nm and 427.8-nm auroral images. Maximum and minimum erros and errors at the center position of the reconstruction region are listed. The errors were calculated as (Result – Model) / Model.

|  | Maximum error | Minimum error | Center position error |
|---|---|---|---|
| $Q_0$ (427.8 nm) | 10% | –24% | –7% |
| $Q_0$ (557.7 & 427.8 nm) | 11% | –11% | 0.7% |
| $E_0$ (427.8 nm) | 72% | –19% | 47% |
| $E_0$ (557.7 & 427.8 nm) | 29% | –25% | 16% |

energy (Figure 5b). Second, the 3-D distributions of the volume emission rate at wavelengths of 427.8 nm and 557.7 nm were derived using the GLOW model. Third, pseudo auroral images were obtained by integrating the 3-D volume emission rates from the various observation points. Random noises from a normal distribution with a mean value of 0 and standard deviation determined from observed auroral images were added to the pseudo images. Fourth, the total energy flux and average energy were reconstructed using only 427.8-nm images (Figures 5c–d) and the using both 557.7 and 427.8 nm images (Figures 5e–f). Finally, relative errors between the reconstructed and modeled total energy flux and between the reconstructed and modeled average energy were calculated as Error = (Result – Model) / Model (Figures 5i–j). The total energy flux and average energy were underestimated in the northwest and overestimated in the southeast, but the use of auroral images at 557.7 nm and 427.8 nm showed a decrease in error compared to the use of only the 427.8 nm image. Maximum, minimum, and center position values for the total energy flux and average energy are listed in Table 2.

## 3 Results

### 3.1 G-ACT reconstruction results

The precipitating electron fluxes and volume emission rates at wavelengths of 557.7 nm and 427.8 nm were reconstructed from auroral images using G-ACT. Figure 6 depicts the total energy flux and average energy of the reconstructed precipitating electron flux. The total energy flux reached its maximum value of ~2.2–2.3 mW m$^{-2}$ near the center part of the PsA patch from 00:53:34 to 00:53:38 UT. The average energy reached its maximum value of ~62 keV at 00:53:38 UT. Figures 7 and 8 illustrate the 3-D distribution of the reconstructed volume emission rate at wavelengths of 557.7 nm and 427.8 nm, respectively. The peak altitudes of the volume emission rate along the EISCAT radar beam were approximately 94 km for 557.7 nm and 86–88 km for 427.8 nm (Figures 7b, 7d, 8b, and 8d). To validate the G-ACT reconstruction results, the difference between the observed auroral image (Figure 9a) and the pseudo auroral image (Figure 9b) obtained by integrating the reconstructed volume emission rates from the various stations was calculated and is shown in Figure 9c. The observed auroral images were almost perfectly reconstructed except for images at ABK, for which the reconstruction region lay near the edge of the ASC's field of view. One of the reasons for this error was that the ground stations were biased to the





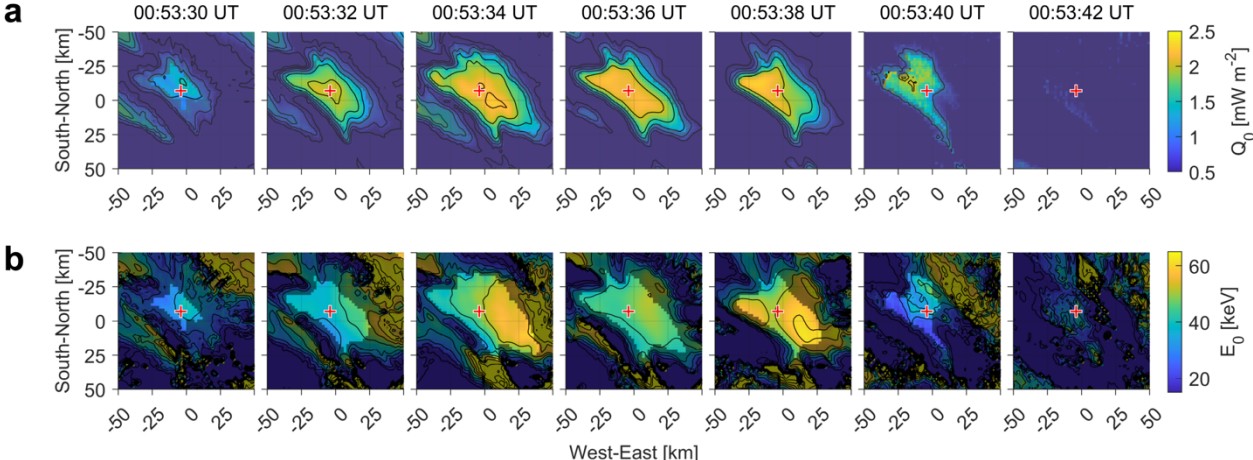

**Figure 6**. **(a)** The total energy flux $Q_0$ and **(b)** average energy $E_0$ of the reconstructed electron flux. The pixels where the total energy flux was less than 1.0 mW m$^{-2}$ are masked by translucent black. Red pluses represent the EISCAT radar observation point.

south of the 12nalysed PsA patch. Therefore, increasing the number of observation points from the north or targeting an auroral structure closer to the centroid of the observation points is expected to mitigate this error.






**Figure 7. (a, c)** The 3-D distribution of the reconstructed volume emission rate (VER) at a wavelength of 557.7 nm viewed from different elevation angles. **(b)** Cross-sections parallel to the magnetic field lines and **(d)** horizontal cross-sections containing peak values on the EISCAT radar beam. Vertical and horizontal dashed lines show the EISCAT radar beam and the altitude of peak volume emission rate, respectively.

**Figure 8.** The reconstructed volume emission rate at a wavelength of 427.8 nm. Figure format is the same as that of Figure 7.




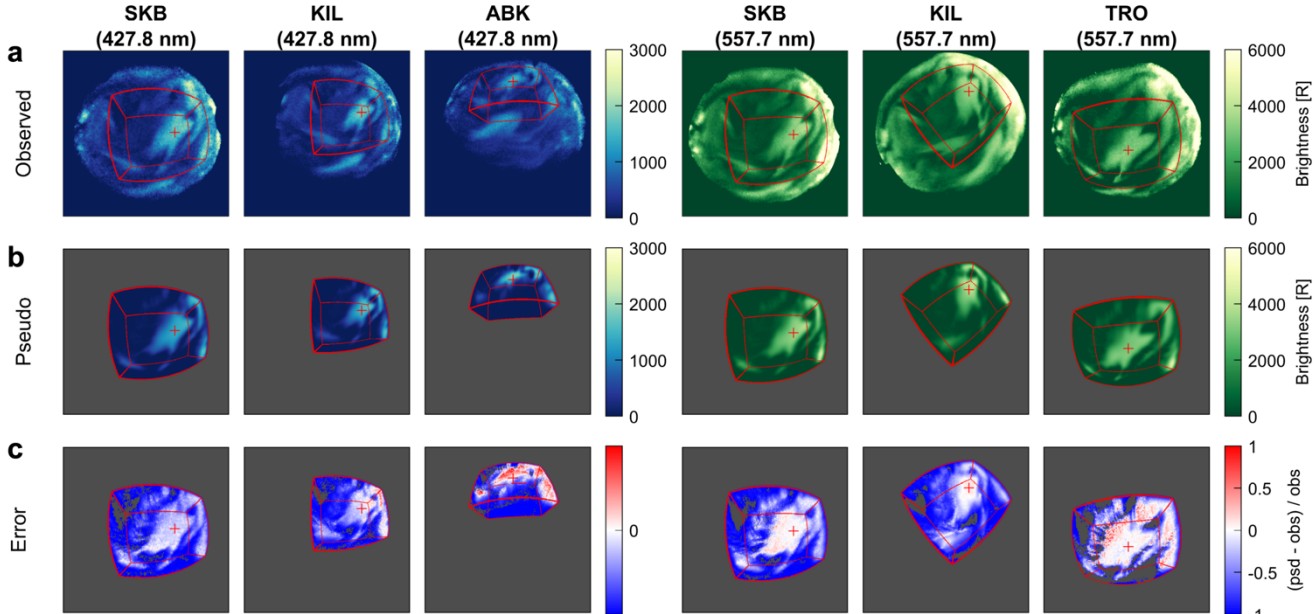

**Figure 9. (a)** Observed auroral images at 00:53:36 UT on February 18, 2018. Red pluses and lines represent the EISCAT radar observation pixel and the reconstruction region boundaries, respectively. **(b)** Pseudo auroral images, calculated from the reconstructed volume emission rate. **(c)** Relative errors, calculated as [(b) – (a)] / a.

## 3.2 Three-dimensional Pedersen and Hall conductivities

The electron density was derived from the reconstructed volume emission rate at 427.8 nm (Figure 8) by solving the continuity equation of the electron density (Eq. (9)) with the Runge–Kutta method, as explained in Section 2.2. Figure 10 displays the reconstructed 3-D electron density. The effective recombination coefficient used was (Vickrey et al., 1982)

$$\alpha_{\text{fit}} = 2.5 \times 10^{-12} \exp(-z/51.2) \ [\text{m}^3 \ \text{s}^{-1}], \tag{14}$$

where $z$ [km] is the altitude. The peak electron density on the EISCAT radar beam occurred at an altitude of almost 98 km (Figs. 10b and 10d). The method of solving the continuity equation of the electron density accounted for time variation and thus was able to show that the electron density remained high even after the brightness of the aurora faded at 00:53:42 UT. To evaluate the reconstruction accuracy of the electron density, the reconstructed altitudinal profile of electron density was

compared with that observed by the EISCAT radar (Figure 11a). In addition to the effective recombination coefficient of Eq. (14), two coefficients were used as the upper and lower bounds on $\alpha_{\text{eff}}$ in Eq. (9) (Semeter and Kamalabadi, 2005), namely,

$$\alpha_{\text{NO}^+} = 4.2 \times 10^{-13} (300/T_{\text{n}})^{0.85} \ [\text{m}^3 \ \text{s}^{-1}], \tag{15}$$

$$\alpha_{\text{O}_2^+} = 1.95 \times 10^{-13} (300/T_{\text{n}})^{0.7} \ [\text{m}^3 \ \text{s}^{-1}], \tag{16}$$

where $T_{\text{n}}$ [K] is the neutral temperature. The observed electron densities lay mostly within the electron density distribution

obtained using the three effective recombination coefficients.



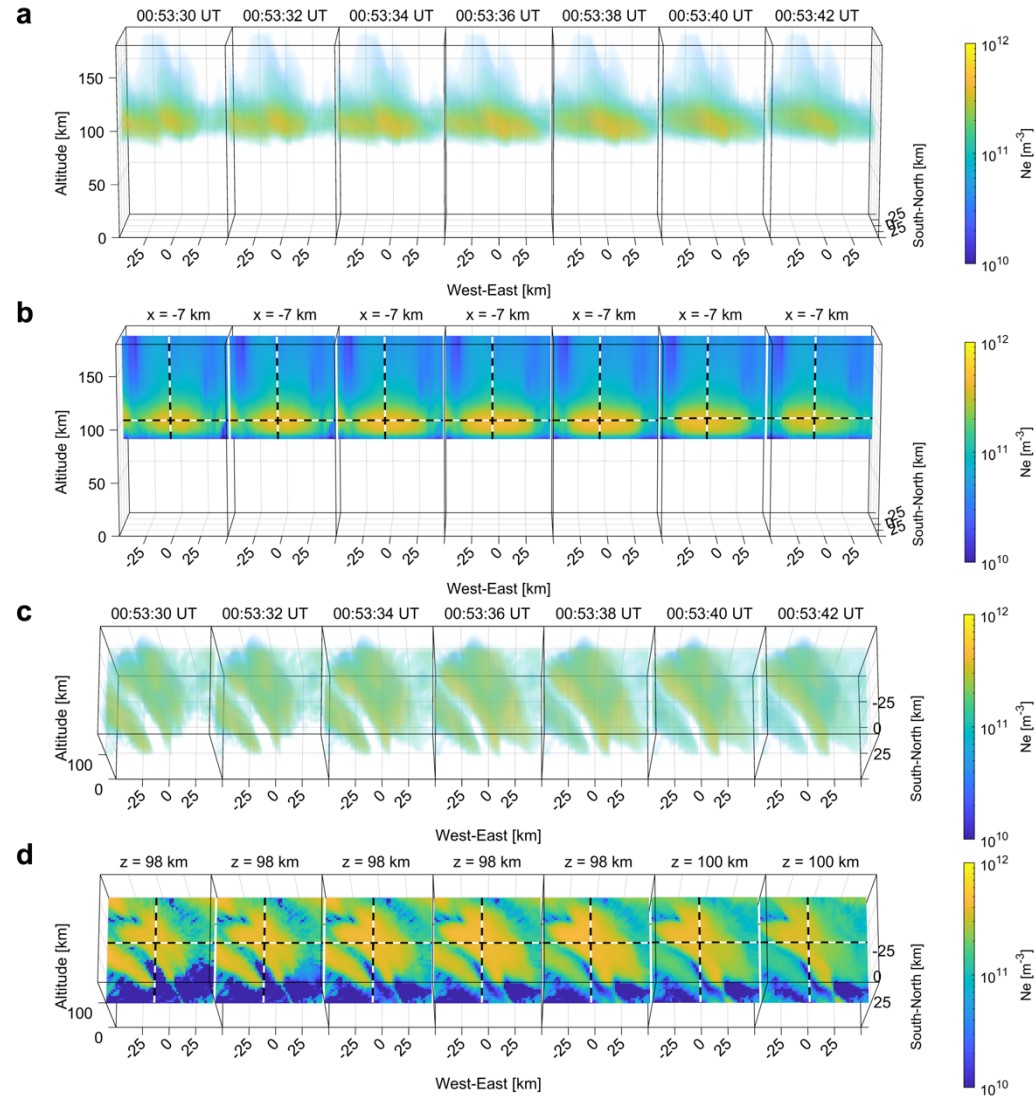

**Figure 10. (a, c)** Reconstructed 3-D electron density ($N_e$) viewed from different elevation angles. **(b)** Cross-sections parallel to the magnetic field lines and **(d)** horizontal cross-sections containing peak values on the EISCAT radar beam. Vertical and horizontal dashed lines represent the EISCAT radar beam and the altitude of peak electron density, respectively.

The Pedersen and Hall conductivities were calculated by substituting the reconstructed electron density (Fig. 10) into Eqs. (1) and (2). Figures 12 and 13 illustrate the 3-D distribution of the reconstructed Hall and Pedersen conductivities. The Hall conductivity reached its maximum value of $1.4 \times 10^{-3}$ S m$^{-1}$ at 94 km altitude, while the Pedersen conductivity reached its maximum value of $2.6 \times 10^{-4}$ S m$^{-1}$ at 116 km altitude. The Pedersen conductivity showed a secondary peak of $9.9 \times 10^{-5}$ S m$^{-1}$ at 86 km altitude. The electron Pedersen conductivity maximum value in the D region was approximately 38% of the ion Pedersen conductivity maximum value in the E region, compared to a figure of 13% in Hosokawa and Ogawa (2010). The



altitude profiles of the reconstructed Hall and Pedersen conductivities were compared with those calculated from the electron densities observed by the EISCAT radar (Figures 11b and 11c). Although the Pedersen conductivity values reconstructed by G-ACT were overestimated in the *D* region compared to those of the EISCAT radar by a factor of 1.2–1.5, the *D*-region

Pedersen conductivity peak values derived from the EISCAT radar were also 25–44% of those in the *E* region, in keeping with the G-ACT results.

**Figure 11.** Altitude profiles of the reconstructed (red lines) and observed (black lines) **(a)** electron density, **(b)** Hall conductivity, and (c) Pedersen conductivity. Error bars represent measurement uncertainties.

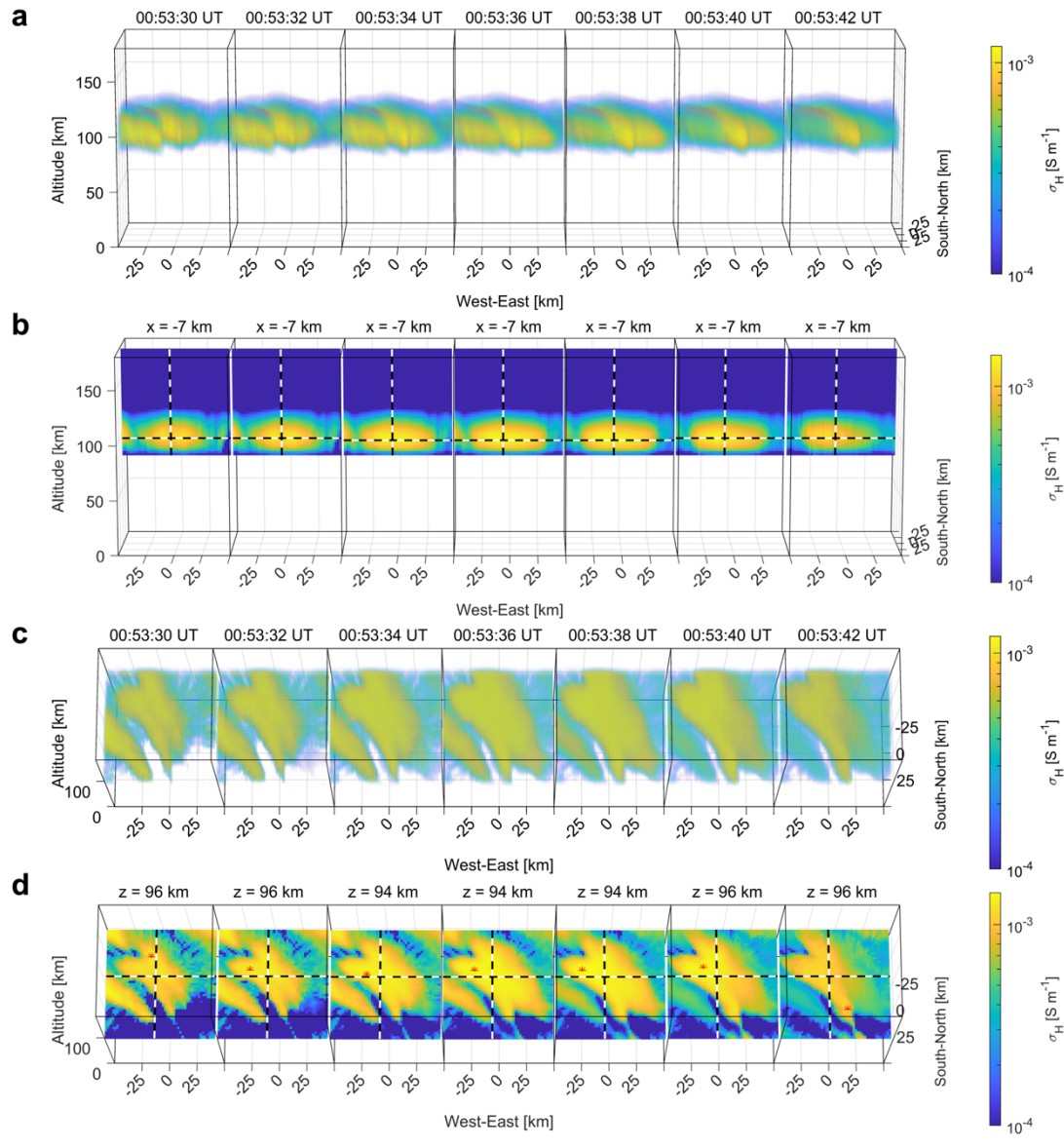

**Figure 12. (a, c)** The 3-D distribution of the reconstructed Hall conductivity. **(b)** Cross-sections parallel to the magnetic field lines and **(d)** horizontal cross-section containing peak values on the EISCAT radar beam. Vertical and horizontal dashed lines show the EISCAT radar beam and the altitude of peak Hall conductivity, respectively. Red asterisks represent peak value positions within the center of the PsA patch.

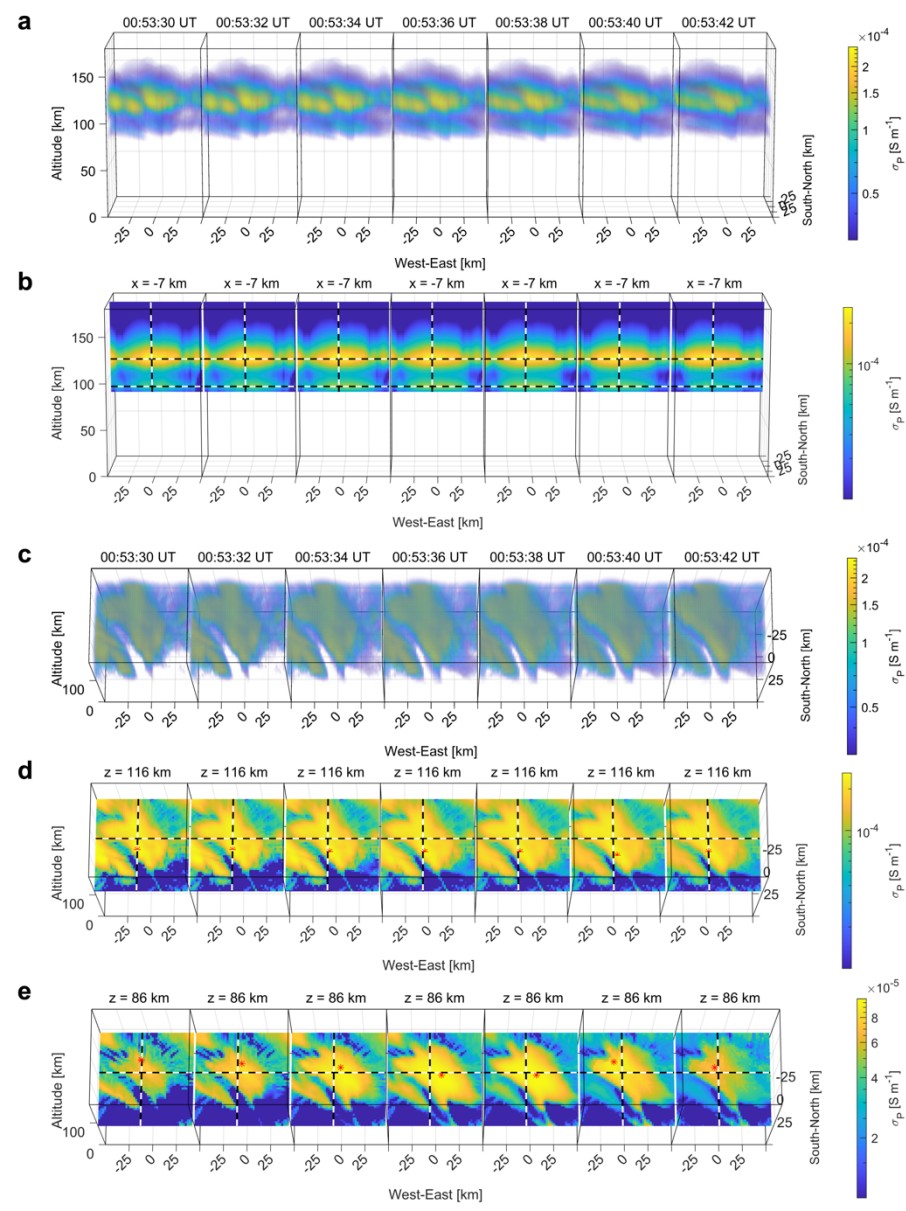

**Figure 13.** The 3-D distribution of the reconstructed Pedersen conductivity. The figure format is the same as that of Figure 12.






## 4 Discussion

The horizontal distribution of the Pedersen conductivity in the $D$ region varied significantly in space and time compared to the $E$ region (Figs. 13d and 13e). The positions of peak values differed between the electron and ion Pedersen layers. These altitudinal differences in the horizontal distribution of the Pedersen conductivity affect the closure of FACs associated with

PsAs. It is expected that in the future, the ionospheric conductivity obtained using the method proposed in this study will be combined with EISCAT_3D data to elucidate how the FACs associated with PsAs are closed in the ionosphere.

3-D electric field data, which will be obtained in the future by EISCAT_3D radar observations, are not available at present. However, the electric field is often assumed to be uniform around PsAs, since PsA patches often drift uniformly with the $\mathbf{E} \times \mathbf{B}$ drift velocity (Hosokawa et al., 2010; Hosokawa and Ogawa, 2010; Oguti et al., 1984; Yang et al., 2015). Therefore,

we evaluated the amount of FAC caused solely by the non-uniform distribution of ionospheric conductivity under the assumption of a uniform electric field. According to current continuity, the FAC, $j_{\parallel}$ [A m$^{-2}$], is equal to the divergence of the height-integrated current perpendicular to the magnetic field lines, $\boldsymbol{J}_{\perp}$ [A m$^{-1}$]. Using the current continuity and ionospheric Ohm's law, the FAC can be written as

$$j_{\parallel} = \boldsymbol{\nabla}_{\perp} \cdot \boldsymbol{J}_{\perp} = (\boldsymbol{\nabla}_{\perp}\Sigma_{\mathrm{P}}) \cdot \boldsymbol{E}_{\perp} + \Sigma_{\mathrm{P}}(\boldsymbol{\nabla}_{\perp} \cdot \boldsymbol{E}_{\perp}) - (\boldsymbol{\nabla}_{\perp}\Sigma_{\mathrm{H}}) \cdot (\boldsymbol{E}_{\perp} \times \boldsymbol{b}) - \Sigma_{\mathrm{H}}\boldsymbol{b} \cdot (\boldsymbol{\nabla}_{\perp} \times \boldsymbol{E}_{\perp}), \qquad (17)$$

where $\Sigma_{\mathrm{P}}$ and $\Sigma_{\mathrm{H}}$ [S] are the height-integrated Pedersen and Hall conductivities, respectively, $\boldsymbol{E}_{\perp}$ [V m$^{-1}$] is the ionospheric electric field perpendicular to the magnetic field, and $\boldsymbol{b}$ is the unit vector of the geomagnetic field. We assumed a uniform southward $\boldsymbol{E}_{\perp}$, allowing the second and fourth terms in Eq. (17) to be ignored. The $\boldsymbol{E}_{\perp}$ amplitude was estimated to be 12.5 mV m$^{-1}$ from the ionospheric convection velocity obtained from Super Dual Auroral Radar Network (SuperDARN) (Greenwald et al., 1995) observations ($\sim 250$ m s$^{-1}$) and assuming $B \approx 50000$ nT. Consequently, the maximum downward

and upward FACs were 69 and 68 μA m$^{-2}$ at the northeast and southwest edges of the PsA patch, respectively. These FACs are approximately ten times larger than those observed by magnetometers onboard satellites (Gillies et al., 2015). Two main factors could cause this overestimation of FACs. One is that the uniform $\boldsymbol{E}_{\perp}$ assumption may not be reasonable. Indeed, higher conductivities drive the polarization of the electric field inside PsA patches (Hosokawa et al., 2010; Takahashi et al., 2019). The 3-D distribution of the ionospheric electric field will be obtained with the EISCAT_3D radar in late 2023,

settling this question. The other factor is the background emission subtraction from the auroral images before conducting G-ACT. Background emission subtraction enabled accurate reconstruction within the PsA patch. However, the conductivity outside the patch will be underestimated if the subtracted background emission contains mainly diffuse auroral emission. On the contrary, if uniform diffuse auroral emission occurs around 150 km altitude, above PsAs, as reported by Brown et al. (1976), then the subtraction is not a problem because the horizontal gradient of electrical conductivity near the PsA patch is

not affected by the subtraction. It is also possible that the PsA patches analyzed in this study actually had such large FACs,



given that their boundaries were sharper than those of Gillies et al. (2015), who derived PsA FACs from satellite observations. The horizontal distribution of the ionospheric conductivity around PsA patches will be investigated by the EISCAT_3D radar.

## 5 Conclusions

In this study, G-ACT was used to reconstruct the 3-D distributions of the Hall and Pedersen conductivities of PsAs in order to elucidate the 3-D structures of ionospheric currents via documentation of the 3-D distribution of ionospheric conductivity. The tomographic results show that the Hall conductivity peaked in the $E$ region (altitudes of 94–96 km) while the Pedersen conductivity peaked in the $E$ region (at 116 km altitude) with a secondary peak in the $D$ region (at 86 km altitude). The electron Pedersen conductivity maximum value in the $D$ region was approximately 38% of the ion Pedersen conductivity

maximum value in the $E$ region; this ratio was nearly triple that reported by Hosokawa and Ogawa (2010). This result suggests that the Pedersen current in the $D$ region caused by high-energy electron precipitation associated with PsAs has an outsized effect on FAC closure in the ionosphere.

Under the assumption of a uniformly distributed ionospheric electric field, derived FAC values near the edges of PsA patches were approximately ten times higher than those reported by satellite observations. This overestimation means that

the ionospheric electric field is not uniform and that the electrical conductivities around PsA patches may be underestimated. In the near future, the 3-D ionospheric conductivity reconstruction using G-ACT proposed in this study will be combined with 3-D observations of ionospheric conductivity and electric field strength by EISCAT_3D radar to elucidate the 3-D ionospheric current structures associated with PsAs.

**Data availability**

The MIRACLE EMCCD camera data from ABK and KIL are available at ***. The auroral images obtained by the four WMI CCD cameras can be obtained at ***. The EISCAT data can be accessed from http://esr.nipr.ac.jp/www/eiscatdata/. The SuperDARN data can be found at ***. (The data used in this study will be prepared for open access when the manuscript is accepted.)

**Video supplement**

We will get a DOI for Video A1 when this manuscript is accepted. Currently, it is available from the following link: https://www.dropbox.com/s/tzy104lannpgewa/Video_A1.mp4?dl=0.



**Author contribution**

Yoshimasa Tanaka developed the G-ACT method and code. Yasunobu Ogawa conducted the EISCAT radar observation and prepared the ionospheric electron density data. Tero Raita maintained the MIRACLE camera network and prepared the auroral images. Mizuki Fukizawa analyzed the data prepared by co-authors and prepared the manuscript with contributions from all co-authors. Keisuke Hosokawa contributed to the discussion and interpretation of the analysis results.

**Competing interests**

At least one of the authors is a member of the editorial board of Annales Geophysicae.

**Acknowledgements**

The first author is a Research Fellow of the Japan Society for the Promotion of Science (JSPS). This study is supported by JSPS KAKENHI Grant Numbers JP17K05672, JP21H01152, and JP23KJ2145. EISCAT is an international association supported by research organizations in China (CRIRP), Finland (SA), Japan (NIPR), Norway (NFR), Sweden (VR), and the United Kingdom (UKRI). We thank Kellinsalmi Mirjam and Carl-Fredrik Enell for maintaining the MIRACLE camera network and data flow. The database construction for the imager data at Skibotn and the EISCAT radar data has been supported by the IUGONET (Inter-university upper atmosphere Global Observation NETwork) project (http://www.iugonet.org/). The authors acknowledge the use of SuperDARN data. SuperDARN is a collection of radars funded by national scientific funding agencies of Australia, Canada, China, France, Italy, Japan, Norway, South Africa, the United Kingdom, and the United States of America.

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
