# Peer review of "Three-dimensional ionospheric conductivity associated with pulsating auroral patches: Reconstruction from ground-based optical observations"

_Annales Geophysicae, 2023_

## Author Response (AR1)

**Response to Referee Comment 1**

We are grateful for the valuable comments and suggestions provided by the Reviewer. We have considered all the comments and suggestions and made appropriate changes to the revised manuscript.

**Comment 1:**

3. Is the presentation clear?

Yes. However, both figures and language can be improved.

Some points in particular that might benefit from expansion or clarification:

In the equations describing the GACT-procedure it might be better to use "model-images" or "modeled images" instead of "pseudo image" for the images calculated from the reconstructed 3-D volume emission rates.

**Reply 1:**

We have changed "pseudo image" to "modeled image" in the revised manuscript.

**Comment 2:**

I would also have **chosen to use** "~" on the modeled images instead of the observations. Further I would use a subscript "g" on the covariance-matrices ($\mathbf{\Sigma}_g$) and set the station-index "j" under the summation-sign ($\sum\limits_j$) to make the distinction between the two clear in equation 6.

**Reply 2:**

Thank you for the suggestion; however, we have already used "~" for observed images in our previous studies (Fukizawa et al., 2022; Tanaka et al., 2011). Thus, we would like to use it in the same manner.

**Comment 3:**

A **couple of sentences** on the forward model from 3-D volume emission-rate to images could help the reader not familiar with this type of work.

**Reply 3:**

We have added sentences in the revised manuscript to explain how to create the modeled image from the 3-D volume emission rate as follows:

"The image brightness $g_i$ at $i$-th pixel in the modeled auroral image was approximated as follows (e.g., Aso et al., 1990; Tanaka et al., 2011):

$$g_i = \frac{c_g(\theta,\phi)}{4\pi} \int L(r,\theta,\phi)dr, \tag{6}$$

where $(r,\theta,\phi)$ are polar coordinates with origin at the center of the camera lens, and $c_g(\theta,\phi)$ is a sensitivity and vignetting factor (Aso et al., 1990)."

**Comment 4:**

For the effective recombination-rates equations the exponential variation corresponds to some altitude-variation of the O2 and NO ion-mixing-ratio. Is that consistent with the results from the GLOW-runs? Could you mention how well this altitude-variation is consistent with what is expected for pulsating auroras?

**Reply 4:**

We calculated the effective recombination rate using the GLOW model. The auroral electron Maxwellian flux and characteristic energy, which are input values for GLOW, were determined from our reconstruction result of the pulsating aurora (Figure 6). It was assumed that they were 2.5 mW m$^{-2}$ and 30 keV. The calculated effective recombination coefficient was included in the range between Eqs. (16) and (17). Therefore, it is reasonable to use Eqs. (16) and (17) for pulsating auroras. Conversely, Eq. (15) is beyond Eq. (16) below ~90-km altitude, where the Pedersen conductivity had a small peak. In these altitudes, Eq. (15) is appropriate to calculate the Pedersen and Hall conductivities because the electron density calculated using Eq. (15) most closely matches the electron density observed by the EISCAT radar (Figure 11a). Therefore, the Pedersen and Hall conductivities calculated using Eq. (15) are shown in Figures 12 and 13, respectively. These clarifications have been incorporated in our revised manuscript.

**Comment 5:**

Also since the simplified electron continuity-equation is non-linear, the effect of back-ground removal on contribution to E-region ionization from the diffuse aurora should be explained. If the diffuse aurora is so much softer it should be possible to get an indication from the 427.8/557.7 intensiy-ratios?

**Reply 5:**

We have added the following sentences in lines 388–401 in the manuscript:

"The peak altitude of auroral emissions or the characteristic energy of precipitating electrons can be estimated from the emission intensity ratio of 557.7–427.8 nm (Rees and Luckey, 1974; Steele and Mcewen, 1990). The intensity ratio of the removed background emission at 557.7/427.8 was 2.8. This ratio indicates that the characteristic energy of the precipitating electrons was 1.6–4.0 keV (Figure 4 in Rees and Luckey (1974)), ~1.5–5 keV (Figure 9 in Steele and Mcewen (1990)), or ~20 keV using the GLOW model. Thus, the characteristic energy derived from the 557.7/427.8 ratio depends on the models. The background emission intensity subtracted in our analysis was about 30% of the observed pulsating auroral emission. Therefore, the volume emission rate of the background radiation was calculated using the GLOW model and multiplied by a constant so that the ratio of the linearly integrated volume emissivity in the altitude direction is 30% of the volume emission rate of the reconstructed pulsating aurora. Then, the electron density was derived using the electron continuity equation by adding the calculated background emission to the reconstructed volume emission rate. We then examined how much the electron density is underestimated at altitudes of 86, 96, and 116 km, where the Pedersen and Hall conductivities show peak values. For a characteristic energy of 1 keV, the electron density was underestimated by ~30-40% at an altitude of 116 km and remained the same at altitudes of 86 km and 96 km. When the characteristic energy was 20 keV, the underestimation was ~10% at all three altitudes."

**Comment 6:**

The authors might consider shifting the time-frame to span more of a transition between pulsation-off and pulsation-on, either to include more of an on-set or end of a pulsation - this to illustrate the importance of the time-resolution optical observations can contribute with.

**Reply 6:**

We conducted a special experiment using all-sky cameras and the EISCAT radar in February 2018. Owing to the lack of clouds at all four stations and the detection of the pulsating auroral patch at the EISCAT radar observation site, the time specified in the manuscript is optimal for analysis. Future research will focus on pulsating auroras with increased pulsation.

**Comment 7:**

Is the size of each figure adequate to the quantity of data it contains?

Some improvements would help the paper to reach its full potential. Below follows a few suggestions. Many figures contain many panels, which makes the data presented in each individual panel a bit difficult to see. My suggestions are to:

1, Remove the panel-labelings 'a', 'b' etc they eat space, instead the authors should consider making the figures the full width of the page.

**Reply 7:**

We have removed the panel labeling from Figure 2.

**Comment 8:**

2, For figure 2, maybe the time period displayed could be shortened to 00:45 - 01:20 and it might be possible to remove the wavelength-labels and change the intensity-unit from R to kR to give the data more space.

**Reply 8:**

As suggested, we have shortened the time period in Figure 2 to 00:45–01:20 UT and changed the intensity unit from R to kR.

**Comment 9:**

3, In figure 6, perhaps it would be more clear to mark the low-intensity regions with hatching instead of the dark-shading.

**Reply 9:**

We have changed the low-intensity regions with white cross lines.

**Comment 10:**

4, In the figures with volume-renderings (7, 8, 10, 12 and 13) it should be clearer to make standard 2-D plots of the horizontal and vertical cuts, this would also give the data more space.

**Reply 10:**

As suggested, we have changed 3-D slice plots to standard 2-D plots.

**Comment 11:**

5, In figure 9, it might be nicer to zoom-in on the reconstructed regions in the second and third rows of panels.

**Reply 11:**

The second and third rows of panels in Figure 9 have been zoomed in the revised manuscript.

**Comment 12:**

6, In figure 1, make lines thicker and state that the map is over northern Fenno-Scandia.

**Reply 12:**

We have made the lines thicker in figure 1 and stated that the map is over Northern Fennoscandia.

**Minor comment/suggestions:**

**Comment 13:**

Throughout: The **equation-numbering are set inconsistently**. Some numbers appear just at the end of the equation, others close to the right margin.

**Reply 13:**

We have put the equation numbers at the end of the equation.

**Comment 14:**

Line 12: midnight-sector to the noon sector - in principal it is possible to go both east and west from the midnight-sector and reach noon. This reviewer cannot come up with an equally short and un-ambiguous phrase, but urge the authors to try.

**Reply 14:**

We have used "in the midnight-morning-noon sector" instead of "from the midnight sector to the noon sector."

**Comment 15:**

Line 20: The Pedersen conductivity... -> A secondary peak in the Pedersen conductivity, due to electron-motion, at 9.9e-5 S/m appears at 86 km of altitude.

**Reply 15:**

As suggested by the reviewer, we have revised the above sentence in the revised manuscript.

**Comment 16:**

Line 33: PsAs are classified into three types -> PsAs are typically grouped into three classes

**Reply 16:**

We have revised the above sentence as suggested in the revised manuscript.

**Comment 17:**

Line 52: The 3-D distribution... -> In the future the 3-D distribution...

**Reply 17:**

We have revised the above sentence as suggested by the reviewer.

**Comment 18:**

Line 84: ...is the gray level... -> ...is the image brightness... or ...is the pixel photon count...

**Reply 18:**

We have revised "the gray level" to "the image brightness."

**Comment 19:**

Line 85: \nabla^2 f is the second-order derivative - here you need to expand on how you weigh the derivatives in the spatial and energy-dimensions.

**Reply 19:**

The weights for derivatives in space and energy were set to 1. We have mentioned it in the revised manuscript.

**Comment 20:**

Line 98: The subscript j is ian -> The subscript j is an

**Reply 20:**

We have revised this in the manuscript.

**Comment 21:**

Line 111: The fontsize of this equation seems smaller.

**Reply 21:**

The font size is according to the ANGEO's word template. Hence, we have retained it.

**Comment 22:**

Caption to figure 2: ...sliced along the magnetic latitude (MLAT) - does this correspond to an East-West aligned cut? If so make that more explicitly clear since most keograms are made with North-South aligned cuts.

**Reply 22:**

It is North–South aligned cuts. So, we have it to "…sliced along the magnetic longitude…".

**Comment 23:**

Caption to figure 2: ...EISCAT radar observation point <- at what altitude along the radar-beam is the point taken that you project down onto the other images?

**Reply 23:**

The altitude is 100 km, and we have stated it in the caption of Figure 2.

**Comment 24:**

Table 2 perhaps change "Maximum error Minimum error" to "Error-range" and "Centre position error" to "average error".

**Reply 24:**

We have revised "Maximum error Minimum error" to "Error-range" and "Centre position error" to "average error."

**Comment 25:**

Line 195: ...(Figures 5c-d) and the using both... -> ...(Figures 5c-d) and using both...

**Reply 25:**

We have removed "the" from the sentence at line 195 in the revised manuscript.

**Comment 26:**

Line 198: auroral images at 557.7 nm -> auroral images at both  557.7 nm

**Reply 26:**

We have added "both" in the sentence at line 198 in the revised manuscript.

**Comment 27:**

Line 198: ...of only the 427.8 nm image. -> ...of only the 427.8 nm images.

**Reply 27:**

We have corrected "image" from singular to plural.

**References:**
Aso, T., Hashimoto, T., Abe, M., Ono, T. and Ejiri, M.: On the analysis of aurora stereo observations, J. Geomagn. Geoelectr., 42(5), 579–595, doi:10.5636/jgg.42.579, 1990.
Fukizawa, M., Sakanoi, T., Tanaka, Y., Ogawa, Y., Hosokawa, K., Gustavsson, B., Kauristie, K., Kozlovsky, A., Raita, T., Brandstrom, U. and Sergienko, T.: Reconstruction of precipitating electrons and three-dimensional structure of a pulsating auroral patch from monochromatic auroral images obtained from multiple observation points, Ann. Geophys., 40, 475–484, doi:10.5194/angeo-40-475-2022, 2022.
Rees, M. H. and Luckey, D.: Auroral electron energy derived from ratio of spectroscopic emissions 1. Model computations, J. Geophys. Res., 79(34), 5181–5186, doi:10.1029/JA079i034p05181, 1974.

Steele, D. P. and Mcewen, D. J.: Electron auroral excitation efficiencies and intensity ratios, J. Geophys. Res., 95(A7), 10321–10336, doi:10.1029/JA095iA07p10321, 1990.

Tanaka, Y. M., Aso, T., Gustavsson, B., Tanabe, K., Ogawa, Y., Kadokura, A., Miyaoka, H., Sergienko, T., Brändström, U. and Sandahl, I.: Feasibility study on Generalized-Aurora Computed Tomography, Ann. Geophys., 29(3), 551–562, doi:10.5194/angeo-29-551-2011, 2011.

**Response to Referee Comment 2**

We are grateful for the valuable comments and suggestions provided by the Reviewer. We have considered all the comments and suggestions and made appropriate changes to the revised manuscript.

**Comment 1:**

L72–73: Why is equation 3 independent of neutral and ion temperature? Equation A3 of Ieda (2020, doi:10.1029/2019JA027128), for example, shows that the ion-neutral collision frequency in general depends on the relative temperature $(T\_i+T\_n)/2$.

**Reply 1:**

Equation 3 is a simplified approximation for the ion-neutral collision frequency. Ieda (2020) showed that the ionospheric conductivity is underestimated by only 3% when the ion-neutral collision frequency is assumed to depend on the neutral and ion temperatures, which has little impact on our study.

**Comment 2:**

L85–86: What are x and y?

**Reply 2:**

They are axes of the reconstruction region. The x-axis was antiparallel to the horizontal component of the geomagnetic field, whereas the y-axis was parallel eastward. We moved this explanation from the end to the top of the section.

**Comment 3:**

L115: "almost perfectly consistent …" Can the authors make a more quantitative statement about how well the reconstructed e- density matched the EISCAT observations?

**Reply 3:**

We changed the statement in the modified version of the manuscript, which is as follows:

"Using three effective recombination coefficients, Fukizawa et al. (2022) confirmed that the electron density observed by the European Incoherent Scatter (EISCAT) radar is within the calculated electron densities."

**Comment 4:**

L135: Why is flat-field correction done after subtraction? Is this standard procedure? I would have naïvely expected flat-field correction to be done before subtraction.

**Reply 4:**

If an image has a uniform offset, we can conduct flat-field correction before subtracting the offset. However, the images used in our study had a non-uniform offset. Therefore, if we subtract the non-uniform offset after flat-field correction, the image had non-flat-field. This is why we subtracted the offset before flat-field correction. The result of flat-field correction after offset subtraction is shown in Figure 4 of Ogawa et al. (2020a).

**Comment 5:**

L140–141: How are the constants a and b selected?

**Reply 5:**

The constants a and b were determined by the National Institute of Standards and Technology (NIST) traceable 1.9-m integration sphere (Labsphere LMS-760) at NIPR (Ogawa et al., 2020b). We have added this explanation to the revised version of the manuscript.

**Comment 6:**

L154–155: "We eliminated the gray level at pixels whose zenith angle was larger than 80°." Does this mean that you select c and d in equation (12) so that the gray level of pixels with zenith angles ≥ 80° is zero?

**Reply 6:**

No, it does not. We did not use pixel values with zenith angles ≥80° to determine c and d in equation (12).

**Comment 7:**

Line 208: Doesn't the electron density reconstruction also count as validation of the G-ACT method? (Validation of the reconstructed e- density is discussed in §3.2, however, so perhaps this is sufficient?)

**Reply 7:**

Yes, it does. We also think the discussion in section 3.2 is sufficient for reconstruction results about the electron density.

**Comment 8:**

L224: Since it is not at all clear from the citation of Vickrey et al (1982) how this recombination coefficient is derived, I recommend that the authors expand this statement slightly to specify in what manner Vickrey et al (1982) are responsible for equation (14). Did Vickrey et al (1982) derive it? Did they estimate it from measurements? Or did they simply use it on the basis of the work of others, as these authors are presently doing with equation (14)?

**Reply 8:**

We mistook the reference of the recombination coefficient. It is not Vickrey et al. (1982) but Gledhill (1986). We have revised it and added the following sentences to clarify how Gledhil (1986) derived the effective recombination coefficient.

"Gledhil (1986) derived Eq. (15) using the least-squares method for 122 data points of the effective recombination coefficient obtained from 18 previous studies (references in Gledhil, 1986).."

**Comment 9:**

L225: Why is the effective recombination coefficient subscripted with "fit"? In equation (9) the relevant coefficient is subscripted with "eff", so I am confused as to whether the authors wish to make a distinction between alpha_fit and alpha_eff.

**Reply 9:**

They are the same alpha, but we calculated the electron density using three alpha (alpha_fit, alpha_O2+, and alpha_NO+). alpha_fit is the effective recombination coefficient derived by fitting results obtained from previous works.

**Comment 10:**

L231: In line with my comment on L224, what is the purpose of referencing Semeter and Kamalabadi (2005) here? Did they derive or estimate equations (15) and (16)? The authors should describe this, in my opinion. (My experience is that we as a research community are sometimes not careful about checking where we get these various expressions, and are often willing to just use an expression that we find in some paper without looking into its limitations or validity. I am not saying that Fukizawa et al have not been careful, this is just a general observation.)

**Reply 10:**

We cited Semeter and Kamalabadi (2005) because, like them, we used equations (15) and (16) to determine the upper and lower bounds of the effective recombination coefficient, respectively. Equations (15) and (16) are not newly derived by Semeter and Kamalabadi (2005), as they are general formulas found in textbooks.

**Comment 11:**

L253: "The positions of peak values differed between the electron and ion Pedersen layers." What do the authors mean by position? The E-W location, or time, or both?

**Reply 11:**

We meant a position in horizontal planes. We have revised the sentence as follows:

"The peak location in horizontal planes differed between the electron and ion Pedersen layers."

**Comment 12:**

L274: As the authors may be aware, 3D vector measurements with EISCAT_3D are not likely to be available before 2024 or 2025 due to security issues.

**Reply :12**

Thank you for highlighting this. We have revised the sentence as follows:

"The 3-D distribution of the ionospheric electric field will be obtained with the EISCAT_3D radar in the near future, settling this question."

**Comment 13:**

L269-271 and L294-295: Why do the authors refer to their own estimate as an "overestimate"? It's good to be skeptical of one's own estimates, but there are also very good reasons to be skeptical of the estimates given by Gillies et al (2015). Maybe it would help to lengthen this sentence a bit to make it clearer that there are also good reasons for suspecting that the estimate given by Gillies et al (2015) is an underestimate.

**Reply :13**

Thank you for the suggestion. We have added the following sentence in L269-271:

"These FACs are ~10 times larger than those observed by magnetometers onboard satellites (Gillies et al., 2015). It is possible that the PsA patches analyzed in this study had such large FACs because their boundaries were sharper than those of Gillies et al. (2015). Gillies et al. (2015) may also have underestimated the FAC by ~50 μA m$^{-2}$ around 67° magnetic latitude, where the PsA patch was detected in this study, according to Figure 4 in Ritter et al. (2013). If our results are overestimated, mainly there are two main factors."

**Comment 14:**

Table 2: How about adding something like "average absolute error" or "median absolute error"? This would give an indication of the overall degree of improvement when another wavelength and site are included, wouldn't it?

**Reply :14**

I appreciate the suggestion made by the reviewer. We have replaced the center position error with the average absolute error.

**Comment 15:**

Figure 4: In several of the panels in Figure 4b the estimated background exceeds the actually observed brightness. The subtracted brightness should then be negative, shouldn't it? But Figure 4c ("Subtracted") shows that this is apparently not what is done, because none of the brightnesses are negative. So what do the authors do?

**Reply :15**

Yes, the subtracted image had negative values; however, we simply set the color scale to positive. On the other hand, pixel values below 0 were set to 1 before tomographic analysis was performed. We have added this explanation to the revised manuscript.

**Comment 16:**

Figure 6: Sorry if I missed this, why are pixels where the total energy flux < 1 mW/m² masked?

**Reply :16**

The patchy structure of the average energy distribution is difficult to identify (Figure 6b). Therefore, we masked pixels where the total energy flux is <1 mW/m².

**Comment 17:**

Figures 7 and 8: Why do VERs in panels b and d extend to zero, while VERs in panels a and c do not? Are VERs below a certain value in panels a and c made translucent?

**Reply :17**

Yes, the transparency depends on the VER values. The minimum VER is invisible, and the maximum VER is opaque in panels a and c. We have also set the same transparency in panels b and d.

**Comment 18:**

Figure 9c: Why are some regions of the error grayed out? This is especially noticeable for TRO in Figure 9c. I can't see that this is discussed anywhere.

**Reply :18**

In Figure 9c, errors with observed auroral brightness less than 100 are not shown because the relative error (= [(Derived) − (Observed)]/(Observed)) became large, although the absolute difference (= (Derived) – (Observed)) was small.

**Comment 19:**

Figures 10: Similar to Figures 7 and 8, why does the e- density go down to $10^{10}$ m^-3 in panels b and d, but not in panels a and c? This is also seen in Figures 12 and 13.

**Reply :19**

These are the same for Figures 7 and 8. We have also set the same transparency in panels b and d.

**Editorial comments/suggestions 20:**

L89: Would it make more sense to call the "pseudo auroral image" the "modeled auroral image"?

L98: "ian" -> "an"

L131: "adjust" -> "match"

L184: "On the contrary" -> "In contrast"

L213: "12nalysed" -> "analysed"

**Reply 20:**

We appreciate the suggestions and corrections made by the reviewer. We have made all the changes listed above in the revised manuscript.

**References:**

Gledhill, J. A.: The effective recombination coefficient of electrons in the ionosphere between 50 and 150 km, Radio Sci., 21(3), 399–408, doi:10.1029/RS021i003p00399, 1986.

Ogawa, Y., Tanaka, Y., Kadokura, A., Hosokawa, K., Ebihara, Y., Motoba, T., Gustavsson, B., Brändström, U., Sato, Y., Oyama, S., Ozaki, M., Raita, T., Sigernes, F., Nozawa, S., Shiokawa, K., Kosch, M., Kauristie, K., Hall, C., Suzuki, S., Miyoshi, Y., Gerrard, A., Miyaoka, H. and Fujii, R.: Development of low-cost multi-wavelength imager system for studies of aurora and airglow, Polar Sci., 23, doi:10.1016/j.polar.2019.100501, 2020a.

Ogawa, Y., Kadokura, A. and Ejiri, M. K.: Optical calibration system of NIPR for aurora and airglow observations, Polar Sci., 26, doi:10.1016/j.polar.2020.100570, 2020b.

Ritter, P., Lühr, H. and Rauberg, J.: Determining field-aligned currents with the Swarm constellation mission, Earth, Planets Sp., 65(11), 1285–1294, doi:10.5047/eps.2013.09.006, 2013.

Semeter, J. and Kamalabadi, F.: Determination of primary electron spectra from incoherent scatter radar measurements of the auroral E region, Radio Sci., 40(2), n/a-n/a, doi:10.1029/2004RS003042, 2005.

Vickrey, J. F., Vondrak, R. R. and Matthews, S. J.: Energy deposition by precipitating particles and Joule dissipation in the auroral ionosphere, J. Geophys. Res. Sp. Phys., 87(A7), 5184–5196, doi:10.1029/ja087ia07p05184, 1982.